# Mutation resource of Samba Mahsuri revealed the presence of high extent of variations among key traits for rice improvement

**Gopi Potupureddi[1]☯, Vishalakshi Balija[1]☯, Suneel Ballichatla[1]☯, Gokulan C. G.[2]☯, Komal Awalellu[2], Swathi Lekkala[1], Karteek Jallipalli[1], Gayathri M. G.[1], Ershad Mohammad[1], Milton M[1], Srikanth Arutla[1], Rajender Burka[1], Laha Gouri Shankar[3], Padmakumari Ayyangari Phani[3], SubbaRao Lella Venkata[1], Sundaram Raman Meenakshi[1], Viraktamath B. C.[1], Ravindra Babu Vemuri[1], Kranthi Brahma[2], Raju Madnala[2], Hitendra Kumar Patel[2], Ramesh Venkata Sonti[2]\*, Maganti Sheshu Madhav[1]\***

**1** Crop Improvement, ICAR- Indian Institute of Rice Research, Hyderabad, India, **2** CSIR-Centre for Cellular and Molecular Biology, Hyderabad, India, **3** Crop Protection, ICAR- Indian Institute of Rice Research, Hyderabad, India

☯ These authors contributed equally to this work.

\* sheshu24@gmail.com (MSM); sonti@ccmb.res.in (RVS)

**Data Availability Statement:** All relevant data are within the manuscript and its Supporting information files.

## Abstract

To create novel variants for morphological, physiological, and biotic stress tolerance traits, induced mutations were created using Ethyl Methane Sulphonate (EMS) in the background of Samba Mahsuri (BPT 5204), a popular and mega rice variety of India. A population derived from 10, 500 $M_1$ plants and their descendants were phenotyped for a wide range of traits leading to the identification of 124 mutants having variations in key agro-morphological traits, and 106 mutants exhibiting variation for physiological traits. Higher yield is the ultimate goal of crop improvement and we identified 574 mutants having higher yield compared to wild type by having better yield attributing traits. Further, a total of 50 mutants showed better panicle exertion phenotypes as compared to Samba Mahsuri leading to enhancement of yield. Upon rigorous screening for three major biotic stresses, 8 mutants showed enhanced tolerance for yellow stem borer (YSB), and 13 different mutants each showed enhanced tolerance for sheath blight (ShB) and bacterial leaf blight (BLB), respectively. In addition, screening at multiple locations that have diverse field isolates identified 3, 3, and 5 lines for tolerance to ShB, YSB and BLB, respectively. On the whole, 1231 desired mutant lines identified at $M_2$ were forwarded to an advanced generation ($M_5$). PCR based allele mining indicated that the BLB tolerant mutants have a different allele than the reported alleles for well-known genes affecting bacterial blight resistance. Whole genome re-sequencing revealed substantial variation in comparison to Samba Mahsuri. The lines showing enhanced tolerance to important biotic stresses (YSB, ShB and BLB) as well as several economically important traits are unique genetic resources which can be utilized for the identification of novel genes/alleles for different traits. The lines which have better agronomic features can

**Funding:** Council of Scientific and Industrial Research (CSIR), New Delhi, India and ICAR -IIRR. The funders had no role in study design, data collection and analysis, decision to publish, or preparation of the manuscript.

**Competing interests:** NO authors have competing interests.

be used as pre-breeding lines. The entire mutant population is maintained as a national resource for genetic improvement of the rice crop.

## Introduction

Rice is the staple food for almost 56% of the world's population and it is the source of 20% of the world's dietary energy supply. Global rice consumption is projected to increase from 500 million to around 750 million tons by 2050 [1]. Hence, rice production has to be enhanced (50% projected growth) to match the increased requirements of the future. Development of genetic stocks suitable for various agro-climatic zones is being done based on selection among the improved cultivars or through bringing improvements of mega varieties by modifying specific traits. This is often leading to a narrow genetic base which results in low variability and consequently limits the possibilities for recombination and genetic segregation, leading to reduced genetic gains [2]. Therefore, in spite of the development and release of numerous rice varieties and hybrids, the genetic gain has been relatively low [3]. Hence, there is a need to enhance available variation amongst the rice germplasm. Induced mutations are one of the best available options to create variation among the well adapted mega varieties. Mutations induced artificially using physical agents like fast neutron [4], γ-ray [5, 6], ion beam [7] and chemical mutagens such as ethyl methane sulfonate (EMS) [8–10], methyl nitrosourea (MNU) [11], sodium azide (SA) [12, 13] have been employed in rice to create mutations. Chemical or physical mutagens have higher mutation efficiencies as compared to insertions created by T-DNA [14], transposable elements [15, 16] or RNAi [17], TALEN-based gene editing [18], and CRISPR/Cas9 genome editing [19].

Among the mutagens, EMS is being used frequently owing to its production of high frequency (2 to10 mutations/ Mb) of single nucleotide changes (point mutations) by alkylation of specific nucleotides [9, 20] and a relatively small population, ca. 10,000 plants is sufficient to saturate the genome with mutations. According to FAO/IAEA- MVD reports (2020) ([www.mvd.iaea.org](www.mvd.iaea.org)), a total of 828 mutant varieties were developed in rice among them 60 were derived through chemical mutagenesis. Till date using chemical mutagens, 6 mutant resources in diverse *Japonica* cultivars (Nipponbare, Tainung 67, Kitaano, TC65, Yukihikari, Kinmaze, and BRS Querencia), three mutant resources in *indica* cultivars (IR64, Kasalath, and SSBM) and one *aus* variety (Nagina22) were developed [21]. The International Rice Functional Genomics Consortium announced the public availability of more than 200,000 rice mutant lines, which represent mutations in about half of the known functional genes mapped for rice to date [22]. In India, one EMS mutant resource was developed in the genetic background of Nagina 22 (N22) which is a drought and heat tolerant *aus* rice variety [10]. Using this mutational resource, Shoba et al. [23] identified a herbicide tolerant mutant (HTM-N22.)

Samba Mahsuri (BPT-5204) is a mega rice variety of India released in the year 1984 and is now also grown in other Asian countries. It is a long duration (145–150 days) fine-grain type variety, yielding an average of 5–6 t ha$^{-1}$ with excellent cooking quality and has wide adaptability. However, this variety is susceptible to biotic stresses like sheath blight, blast, bacterial leaf blight (BLB) and insect pests such as yellow stem borer (YSB), brown plant hopper (BPH) and gall midge. In spite of these constraints, it is favored by consumers and farmers in India, because of its excellent cooking and eating quality characteristics. Enhancing genetic variability in Samba Mahsuri would be useful to tackle some of the existing constraints of this variety. Furthermore, the excellent combining ability of this variety [24] will help in transferring such

traits into other varietal backgrounds. Because of the above features, Samba Mahsuri is an ideal genotype to develop a comprehensive mutant population. Currently, mutagenesis coupled with molecular mapping and development of molecular markers has become an important tool with excellent potential for crop improvement [25]. Markers identified through next-generation sequencing technologies (NGS) are an efficient and cost-effective approach to characterize the variants in mutant collections [26]. Mutant resources are being used not only for creating variation but also for novel gene identification and deciphering gene function [27]. To generate variability in the background of Samba Mahsuri, the current research focused on developing mutants using EMS and characterized them for various morphological, physiological, yield traits and biotic stress tolerance so as to use these mutants as donors in rice improvement programs.

## Materials and methods

### Mutagenesis of rice seeds

The rice variety, Samba Mahsuri (Nucleus seed obtained from Bapatla Rice Research station, Andhra Pradesh, India) was used for mutagen treatment. Based on kill curve analysis the seeds were treated in two different concentrations of Ethyl methane sulphonate (EMS) *i.e.*, 1.2% (1–6819) and 0.8% (6820–10500) by M/s Bench Biotech Company, Gujarat, India. The duration of treatment was about 12hrs and the treated seeds were washed thoroughly under running tap water for 6 h to leach out the residual chemical.

### Development of $M_1$ and $M_2$ generations

A total of 10,500 $M_1$ plants were raised from mutagenized ($M_0$) seeds in the fields of M/s Bench Biotech. These $M_1$ plants were protected from out crossing through bagging and harvested individually to obtain the $M_2$ seeds from main panicles. The collected $M_2$ seeds were distributed among the participating centers CSIR-Centre for Cellular and Molecular Biology (CSIR-CCMB) and ICAR-Indian Institute of Rice Research (ICAR-IIRR) for further studies. The $M_2$ seeds were raised at ICAR-IIRR and International Crop Research Institute for Semi-Arid Tropics (ICRISAT), Patancheru, Hyderabad during *kharif* (June-July to October-November) 2013.

### Phenotypic characterization of the Samba Mahsuri mutants

A large subset of 10,500 $M_2$ lines were subjected to various morphological (Plant height, Pink apicules, Albino, Xantha and Coloration), agronomical (panicle types, complete panicle emergence, early flowering, grain types and high yield with high grain number), physiological (strong culm, different types of flag leaf, sterile plant, stay green and shattering) and biotic stresses (sheath blight, bacterial leaf blight and yellow stem borer) studies.

Plant height was measured from the base of the plant to the tip of the tallest panicle. High yielding mutants were phenotyped based on the yield parameters like number of tillers, number of panicles, panicle length (cm), grain number per panicle and yield per plant (g) at maturity [28]. The phenotypic acceptability of mutants was also measured; the mutants having optimum number of tillers at 4–5 leaf stage were selected. The number of days from seeding to grain ripening (95% of grains on panicle are mature) was recorded and maturity duration of the mutants was observed. The lines that matured earlier than the wild type (Samba Mahsuri) were selected as early maturing mutants. Mutants having length of > 2cm of uppermost internode were selected as EUI (Elongated Upper Internode) mutants [29]. The mutants which showed complete emergence of panicle not having any choking in the flag leaf but not showing

the elongated upper internode were considered as complete panicle emergence (CPE) mutants. The grain lengths were measured by using a digital Vernier caliper (M/s Mitutoyo, USA) and classified based on length and breadth of grain [30]. For the identification of strong culm mutants, the diameter and physical strength of the culm were taken into account. Culm diameter was measured using a digital sliding Vernier caliper in the field from the plants at a height of 30cm above ground level [31], while the physical strength of the culms was measured with a 'prostrate tester' (DIK-7401 Daiki Rika Kogyo co Ltd, Tokyo, Japan) where the plant was pushed at an angle of 45˚ to calculate the pushing resistance [32].

In each generation, the mutants with the trait of interest were identified and advanced to the next generations in ear to row method. The whole mutant population was advanced up to $M_4$ generation considering the diversity. Later on, from $M_5$ generation, the advancement of mutants was done based on yield and uniformity. The data for the above-mentioned traits were recorded at three different crop growth stages *i.e.*, seedling, vegetative, and reproductive stages following DUS guidelines [10].

### Screening of mutant lines for biotic stresses

Screening for Yellow Stem Borer (YSB) resistance/susceptibility was done in a phased manner at both vegetative and reproductive phases by augmenting the natural pest infestation through artificial releases [33]. The susceptibility and resistance of the mutant lines was determined based on the Standard Evaluation Scale (SES) scale, IRRI [34]. The varieties, Pusa Basmati-1, and Samba Mahsuri were used as susceptible checks, while W1263 was the tolerant check for YSB screening. The selected lines were further evaluated in multi location stem borer screening trial under AICRIP (All India Coordinated Rice Improvement Programme)- Entomology programme for two seasons (2016 and 2017).

Screening for sheath blight tolerance/susceptibility of the mutant population was done at two locations *viz.*, ICAR-IIRR and ICRISAT farms using a highly virulent isolate of rice sheath blight pathogen *R. solani* (WGL-12 isolate) as well as hotspot locations of India using virulent isolate of that location. Artificial inoculation of the isolate was done in colonized typha stem bits and culture prepared according to the procedure given by Bhaktavatsalam et al. [35]. Varieties TN1 and Samba Mahsuri (Wild type) were used as susceptible checks, while 'Tetep' was used as a tolerant check for sheath blight screening. Observations were recorded after 20 days of inoculation and scored as per IRRI-SES scale (Standard Evaluation System) [36]. Further, the selected lines which showed tolerance to sheath blight under field and laboratory conditions were screened with 10 diverse *R. solani* isolates (*viz.*, TN14-1, RNR 13-F, TTB-1, WGL-12-1, Gosaba-1, Kaual, PNT, Jamalpur- Bangar, and Imphal-1) collected from various hotspot regions of India. The three best mutant lines were again screened at four hotspot locations (Kaul, Pantnagar, Chinsura and MCP) in India for further confirmation of sheath blight tolerance.

Similarly, the mutagenized population was evaluated for bacterial leaf blight (BLB) under artificial conditions following the clip-inoculation method [37]. The lesion lengths were recorded after 15 days of infection and scored according to SES scale, IRRI [34, 38]. For BB screening, the varieties TN1 and Samba Mahsuri (wild type parent) were used as susceptible checks, Improved Samba Mahsuri (a derivative of Samba Mahsuri having *Xa21*, *xa13* and *xa5* genes) [39] was used as the resistant check. The identified resistant mutants were further screened with multiple *Xoo* pathotypes. These pathotypes included LUD-05-1 (IXo-090; compatible for *xa5*, *xa13* and *Xa21*), ADT (IXo 015; compatible for *xa5* and incompatible for *xa13* and *Xa21*), TNK-12-3 (IXo-281; compatible for *xa5*, *xa13* and incompatible for *Xa21*), CHN/J (IXo-027; compatible for *Xa21*, *xa5* and incompatible for *xa13)* and DRR-1 (IXo-020) pathotype (compatible for all the three genes *xa5*, *xa13* and *Xa21).*

## Assessment of BLB tolerant mutants for allelic status of known resistance genes, *Xa21*, *xa13* and *xa5*

The bacterial leaf blight (BLB) tolerant lines were screened for allelic variations among the reported genes for BLB resistance (namely *Xa21*, *xa13* and *xa5*) through PCR based screening technique. The DNA extraction was carried out by modified Cetyl Trimethyl Ammonium Bromide (CTAB) method [40]. The primer details of BLB resistance genes (*Xa21*, *xa13* and *xa5*) are provided as S1 Table. The PCR reaction volume and profile was followed using the protocol of Hajira et al. [41].

## Whole genome re-sequencing of mutants

The genomic DNA was isolated from three genetic stocks (which include the nucleus seed of Samba Mahsuri which was used for mutagenesis along with two other stocks of this variety) as well as eight mutants with different traits using the CTAB method as mentioned. The isolated genomic DNA was checked for quality on an agarose gel and quantity using Qubit 4 fluorometer (Thermo Fisher Scientific, USA). Library preparation and sequencing was outsourced to Nucleome Informatics Private Limited, Hyderabad, India. In short, sequencing libraries were prepared using Truseq Nano DNA HT Sample Preparation Kit (Illumina, USA) and qualified libraries were sequenced on Illumina HiSeq2500 platform to obtain 2x150 bp reads at approximately 40X coverage.

The obtained high-quality reads from the three genetic stocks of Samba Mahsuri were used for constructing the Samba Mahsuri reference genome using the consensus calling approach described previously [42]. Briefly, the sequenced reads were aligned using *bwa mem* (v0.7.17; http://bio-bwa.sourceforge.net/bwa.shtml) to the *indica* reference genome, R498 [43]. Using *Samtools* (v0.1.20; http://www.htslib.org/doc/samtools.html) the bam files were sorted, and deduplicated. The final bam files were used for variant calling with reference to R498 genome using *freebayes* (v1.0.0) [44]. The variants were filtered using *VCF tools* (v0.1.15) [45] and bash scripts to retain only homozygous SNPs supported by a minimum of 10 reads and with minimum quality score of 30.

All the Samba Mahsuri sequences were processed in the same way and finally all the SNPs were merged to obtain a single VCF file. Using *BCF tools* (v1.7; http://samtools.github.io/bcftools/bcftools.html) *consensus*, the SNPs were replaced in the R498 reference genome to obtain a Samba Mahsuri reference genome.

The Samba Mahsuri reference genome was used for aligning the mutant sequence reads and variant calling using the same pipeline as discussed above. After obtaining the filtered SNPs, only the GC to AT type transitions were retained for further analyses. Custom scripts were used for identifying the chromosome-wise distribution of the SNPs. Dissimilarity index was calculated by dividing the total number of GC to AT type homozygous SNPs by the total number of bases in the reference genome (i.e., 390, 983, 850). *Tassel* software was used for obtaining the relatedness dendrogram [46]. SnpEff (v4.3t) [47] was used for annotating the SNPs.

## Statistical analysis

The grain number and yield data of mutants and wild type were subjected to statistical analysis to determine the significant variation among the mutants and wild type at $M_5$ generation by using Opstat (http://14.139.232.166 › opstat) program for the estimation of analysis of variance (ANOVA), standard deviation and Coefficient of Variation (CV). Least significant difference (LSD) was determined by using XLSTAT software (https://www.xlstat.com/en/).

## Results

A total of 10,500 mutants ($M_1$) were generated upon treatment of Samba Mahsuri with EMS mutagen in two different concentrations. Phenotypic observations at $M_2$ generation for various morphological, physiological, and yield parameters identified 1231 putative mutants (~ 12% of population) showing promising variation for the traits of interest (Table 1; S2–S7 Tables). These mutants were categorized for different traits, among those, the lines showing variation in the yield contributing traits occupied the major proportion (43%), followed by physiological and panicle emergence mutants (15% each) and the panicle type and morphological trait mutants (Fig 1). The selected mutants were subsequently advanced to $M_3$ and $M_4$ generations wherein 418 and 276 mutants showed stable inheritance of variations, respectively (S2–S7 Tables). Considering the traits of interest, 180 mutants were selected and advanced to $M_5$ generation for further studies (S2–S7 Tables).

Upon screening of mutagenic population in $M_2$ generation for three biotic stresses (YSB, ShB, and BLB), 1453 promising mutant lines were identified, of which ShB mutants occupied the major proportion (55%). These mutants were advanced to the subsequent generations and a total of 34 stable mutants were recovered in $M_5$ generation for the three biotic stresses (S8–S10 Tables).

### Variations in the morphological traits

Around (0.79%, 83 mutants) of the total population showed variation in various morphological traits (viz., tall, dwarf, grassy bonsai and pink apiculus) (Fig 2A & 2B). Among these, mutants having tallness (130-155cm) were the highest proportion (51%, 43 mutants) of the total morphological traits, followed by dwarf mutants (43.3%; 36 mutants) with the height being less than 100cm. We also identified one extreme dwarf mutant (bonsai type having a shoot length of 35cm). This mutant had less number of internodes (n = 3) and very narrow internodal length with grassy appearance. The details of the traits and the number of mutants observed in each generation are given in Table 1 and S2 Table.

### Variations in the physiological traits

The $M_2$ mutants were characterized for various physiological traits viz., flag leaf variation, leaf variations, albino, xantha, coloration, strong culm, stay green, early maturity, shattering and sterility. Of the total population, 1.8% mutants (191) showed variations in the physiological traits. Among these, strong culm that imparts lodging resistance is an important trait and we identified 26 mutants (0.24%) with culm strength ranging between 26–32 N/m2, while the wild type showed culm strength of 25N/m2 (Fig 2C). Early maturing (108–132 days) mutants were observed at a frequency of 0.69% (73 mutants) (Fig 2D). Stay-green in the post-anthesis period is found to be an efficient drought tolerance trait and we obtained 4 stay green mutants (0.03%) in the $M_2$ generation. We also obtained a good frequency (0.26%) of sterile plants; however, these could not be maintained further due to non-uniform segregation (S3 Table).

### Variations in panicle types

Panicle type variants (sparse, long, and dense) were observed at a proportion of 0.8% (85) of the total $M_2$ population. Among these, long panicle (30-32cms) type mutants were the highest with 0.47% (50 mutants) (Fig 2E). The dense panicle mutants recorded 300–350 seeds per plant, while the wild type had only 180–200 seeds (Fig 2F; S4 Table).

**Table 1. Number of mutants exhibiting agronomically important traits from $M_2$ to $M_5$ Generation.**

| S. No. | Character | $M_2$ | $M_3$ | $M_4$ | $M_5$ | | |
|---|---|---|---|---|---|---|---|
| | | | | | | Range | |
| | | | | | | Mutant | Wild type |
| I. | **Morphological Characters** | | | | | | |
| i. | Tall plant | 43 | 12 | 10 | 7 | 154-139cm | 97.5cm |
| ii. | Dwarf | 36 | 4 | 2 | 1 | 90cm | 97.5cm |
| iii. | Bonsai (grassy) | 1 | 1 | 1 | 1 | 35cm | 97.5cm |
| iv. | Pink apicules | 3 | 3 | 1 | 1 | Present | Absent |
| | **Total (I)** | **83** | **20** | **14** | **10** | | |
| II. | **Physiological Characters** | | | | | | |
| i. | Flag leaf variations (Broad & Long) | 28 | 18 | 15 | 9 | L- 45.5-48.5cm; W- 2.2–3.1cm | L- 28.2cm W- 1.3cm |
| ii. | Leaf variations (Narrow) | 7 | 4 | 4 | 1 | L- 25.3cm W-0.8cm | Absent |
| iii. | Albino | 5 | - | - | - | Present | Absent |
| iv. | Xantha | 5 | - | - | - | Present | Absent |
| v. | Coloration | 11 | - | - | - | Present | Absent |
| vi. | Strong culm (Culm strength) | 26 | 17 | 17 | 17 | 26-32N/m$^2$ | 25N/m$^2$ |
| vii. | Maturity (Early flowering) | 73 | 36 | 36 | 36 | 108-132days | 135days |
| viii. | Stay green | 4 | 3 | 2 | 2 | Present | Absent |
| ix. | Sterile plant | 28 | - | - | - | Present | Absent |
| x. | Shattering | 4 | 3 | 2 | 2 | Present | Absent |
| | **Total (II)** | **191** | **81** | **76** | **67** | | |
| III. | **Panicle Type** | | | | | | |
| i. | Sparse Panicle | 5 | 3 | 2 | 2 | Present | Absent |
| ii. | Compact/Dense Panicle | 30 | 13 | 8 | 5 | Present | Absent |
| iii. | Long Panicle | 50 | 32 | 28 | 15 | 30-32cm | 15.6cm |
| | **Total (III)** | **85** | **48** | **38** | **22** | | |
| IV. | **Panicle Emergence** | | | | | | |
| i. | Elongation of Upper Internode (EUI) | 27 | 11 | 11 | 9 | 7-26cm | Absent |
| ii. | Complete panicle emergence | 155 | 31 | 22 | 13 | 2.0–2.5cm | Absent |
| | **Total (IV)** | **182** | **42** | **33** | **22** | | |
| V. | **Yield** | | | | | | |
| i. | More number of tillers | 13 | 9 | 8 | 5 | 15–26 | 12 |
| ii. | More productive tillers & better grain filling | 484 | 120 | 22 | 8 | 13–25 | 12 |
| iii. | High grain number | 15 | 3 | 2 | 2 | 221–306 g/p | 200g/p |
| iv. | Phenotypic acceptability | 20 | 8 | 4 | 2 | Present | Absent |
| | **Total (V)** | **532** | **140** | **36** | **17** | | |
| VI. | **Grain types** | | | | | | |
| i. | Long slender grain without awns | 32 | 18 | 18 | 7 | L-12.6-13.0mm B- 2.2–2.5mm | Absent |
| ii. | Long slender grain with awns | 11 | 11 | 10 | 3 | L-24-29mm B- 2.2–2.5mm | Absent |
| iii. | Long bold grain with awns | 2 | 2 | 1 | 1 | L-19mm B-3.2mm | Absent |

(*Continued*)

**Table 1.** (Continued)

| S. No. | Character | M₂ | M₃ | M₄ | M₅ | | |
|---|---|---|---|---|---|---|---|
| | | | | | | Range | |
| | | | | | | Mutant | Wild type |
| iv. | Long bold grain without awns | 27 | 16 | 14 | 10 | L-10.2-12.2mm B-3.1–3.5mm | Absent |
| v. | Medium slender grain with awns | 4 | 4 | 3 | 1 | L-9.8mm B-2.3mm | Absent |
| vi. | Medium bold grain with awns | 2 | 2 | 1 | 1 | L-9.9mm B-3.4mm | Absent |
| vii. | Medium bold grain without awns | 38 | 22 | 20 | 15 | L-8.9–9.0mm B-3.3–3.5mm | Absent |
| viii. | Short bold | 13 | 4 | 4 | 2 | L-4.8–5.0mm B-3.2–3.5mm | Absent |
| ix. | Short slender | 22 | 5 | 5 | 2 | L-5.1–5.4mm B-2.3–2.4mm | Absent |
| x. | Sickle shape | 7 | 3 | 3 | - | Present | Absent |
| | **Total (VI)** | **158** | **87** | **79** | **42** | | |
| | **Total (I+II+III+IV+V+VI)** | **1231** | **418** | **276** | **180** | | |

L-Length; B- Breadth; W- Width; g/p-grains per panicle; * indicates the only selected and non segregating mutants

## Variations in panicle emergence

Complete panicle emergence (CPE) and Elongated Upper Internode (EUI) are two major traits that were studied under the category of panicle emergence. In this category, we identified 182 mutants (1.72% of the population) of which CPE mutants were more in number (1.47% of the population) and the EUI mutants were less frequent (0.25% of the population). The CPE mutants exhibited 2.0 to 2.5cm of panicle exertion and thus no panicle choking (Fig 2G)

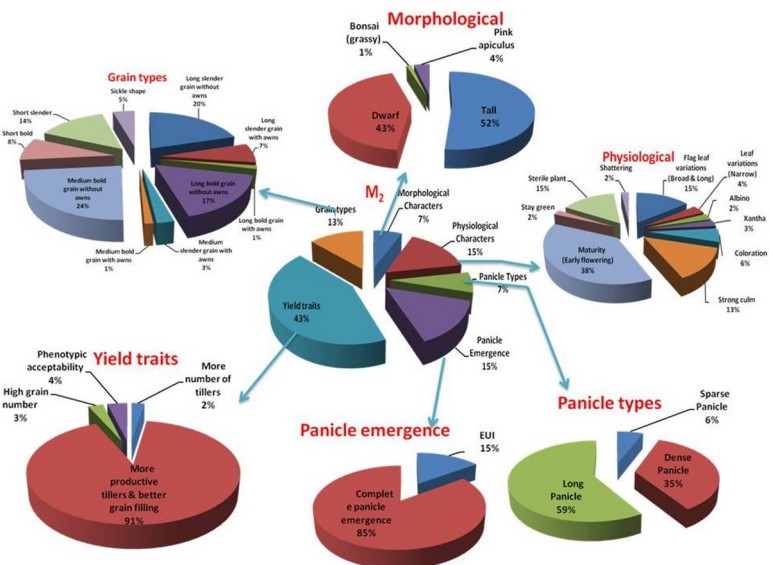

**Fig 1. Distribution of mutants for various agro-morphological traits in M₂ generation.**

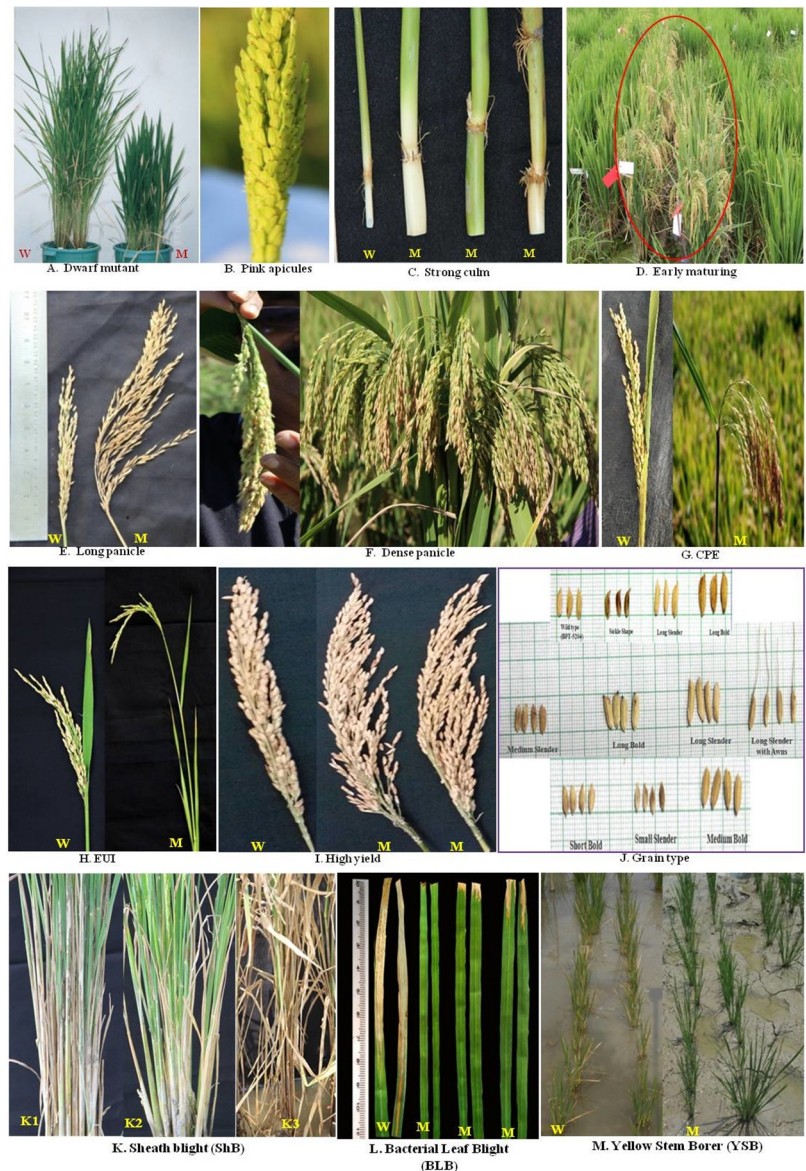

**Fig 2. A—M: Variants for different traits in the mutagenized population of Samba Mahsuri; W- Wild type Samba Mahsuri; M- Mutant; K1- Wild type with score 9; K2- Resistant mutant (SB 6) with score 1 and K3- Susceptible check TN 1 with score 9.**

whereas 10–20% panicle chocking was observed in wild type. The EUI mutants exhibited a length of 7-26cm for the upper most internode (Fig 2H) (S5 Table).

## Variations in yield and yield attributing traits

The yield attributing traits included more number of tillers, number of productive tillers and better grain filling, grain number and phenotypic acceptability (PA). In the $M_2$ generation, the mutants in yield traits were the most frequent; i.e. a total of 532 mutants representing 5.06% of the total mutant population (Fig 2I). Among these traits, number of productive tillers and better grain filling occupied the major proportion with 4.06% (484 mutants), while more number of tillers (15–26 tillers) occupied the least with a proportion of 0.12%. Overall phenotypic

acceptability of the mutant lines was also recorded and we obtained a frequency of 0.19% (20 mutants) in the $M_2$ generation (S6 Table).

## Variations in the grain types

The wild type has medium slender grain type, however upon treatment with EMS we observed a wide range of grain types (Table 1). The grain type mutants constituted 1.5% (158) of the total $M_2$ population, of which medium bold grain without awns were the highest frequency (0.36%), followed by long slender grains without awns (0.34%). Interestingly, we also observed seven (0.06%) sickle shaped seed mutants. The types of variants obtained are indicated in Fig 2J. All these mutants were forwarded to $M_5$ generation, however in subsequent generations we focused on medium slender and long slender grain types, as these are the commercially preferred grain types (S7 Table).

## Biotic stress variation among the mutants

The mutant population that had been generated was large and screening such a large population at one time for various biotic stresses in the same season was not practical. For ease of operations, 2500 $M_2$ families each were screened in two seasons for all three biotic stresses. While screening in the second season, the selected tolerant mutants from the first season were carried forward for rescreening in the next generation along with the base mutant population. A high frequency (13.83%) of the total population showed tolerance for at least one of the three biotic stresses at $M_2$ generation. But after rescreening till $M_5$ generation, the frequency of tolerant lines reduced from 13.83% to 0.33%. At the $M_2$ generation, sheath blight tolerant mutants were the most frequent (7.62%) followed by the bacterial leaf blight tolerant lines (4.59%) and the YSB tolerant lines (1.61%) (S8–S10 Tables). The reduction in frequency of tolerant mutants following rescreening indicates that there were a large number of false positives possibly due to lines that escaped infection. The details of the putative tolerant mutants obtained in each generation for various biotic stresses are given in Table 2 and the visual symptoms are given in Fig 2K–2M.

## Characterization of mutants at multi locations for biotic stress

Among 13 sheath blight mutants, three mutant lines (SB-5, SB-6 and SB-8) showed high level of resistance with a mean-scores of 2.84, 3.24 and 3.28 whereas BPT 5204 (wild type), TN-1 and resistant control (Tetep) showed score of 9, 9 and 4.45, respectively (S11 Table). Further, these three best mutant lines revealed the similar level of tolerance at four hotspot locations of India. This indicates that these three lines are extremely good sources of tolerance for sheath blight. One of the sheath blight mutant (Sb-5) is registered recently with National bureau of plant genetic resources (NBPGR, New Delhi) as INGR 20080 (available at http://www.nbpgr.ernet.in:8080/registration/ApplicationList.aspx). A total of 13 lines showed consistent resistance reaction for bacterial blight under field conditions. Among them 5 (BB-42, BB-49,

**Table 2. Number of mutants exhibiting resistance to biotic stresses (YSB, BLB and Sheath Blight) from $M_2$ to $M_5$ generations.**

| S. No | Trait | $M_2$ | $M_3$ | $M_4$ | $M_5$ |
|---|---|---|---|---|---|
| 1. | Yellow Stem Borer (YSB) | 170 | 30 | 10 | 08 |
| 2. | Bacterial Leaf Blight (BLB) | 482 | 261 | 40 | 13 |
| 3. | Sheath Blight (SB) | 801 | 620 | 40 | 13 |
| | **Total** | **1453** | **1181** | **90** | **34** |

BB-61, BB-132 and BB-133) lines showed resistance level ranging from mean score 1 to 3 to all the tested pathotypes, indicating that the selected mutants were resistant to multiple isolates of the pathogen in India (S11 Table) whereas the wild type (Samba Mahsuri), resistant (Improved Samba Mahsuri) and susceptible (TN-1) checks exhibited mean scores of 7.4, 1.4, and 9 respectively.

The eight mutants identified to be tolerant to YSB were evaluated under AICRIP trials for two seasons. Among them, two mutants i.e., IIRR-Bio-SB-9/SM-92 (20.48 and 20.04g/hill) and IIRR-Bio-SB-5/SM-48 (17.22 and 20.64 g/hill) were most promising exhibiting tolerance to YSB with more grain yield/ hill compared to wild type (16.92 and 17.15g/hill) and tolerant check, W1263 (16.56 and 19.75g/hill) in two seasons (S11 Table).

## Mutants with multiple trait variations

Most of the mutants exhibited trait-specific variations in comparison with Samba Mahsuri, but a few mutant lines (n = 8) showed variation in multiple traits. For instance, the mutant line TI-24 showed different trait combinations like strong culm, early flowering, medium bold, pink apiculus, complete panicle emergence, broad flag leaf and dense panicle and stay green. Similarly, another mutant TI-21 also exhibited variations in multiple traits viz., strong culm, broad flag leaf and early flowering yet showed medium slender grain like wild type. The details of the mutants which showed multiple trait variations are given in the S12 Table.

The alleles contributing to tolerance in BLB tolerant lines are different from the *Xa21*, *xa13* and *xa5* resistance alleles that have been deployed in resistance breeding programs in India: In recent years, the BLB resistance genes *Xa21*, *xa13* and *xa5* have been frequently used in resistance breeding programs in India. Therefore, the allelic status of these five BLB tolerance lines (BB-42, BB-49, BB-61, BB-132 and BB-133), was assessed with gene specific (*Xa21*, *xa13* and *xa5*) markers. The positive check was Improved Samba Mahsuri (ISM) which has the resistance alleles at all three loci and the susceptible checks were TN1 and Samba Mahsuri (wild type) rice varieties which have the susceptibility alleles at all three loci. The genotyping with gene specific markers indicated that none of the five BLB tolerant lines tested carry the resistance alleles that are present in ISM. Thus, these BLB tolerant lines may have some other genes/alleles that contribute to tolerance.

## Validation of dominant/recessive nature of traits

Selected mutants of strong culm (TI-17) and CPE (CPE-109 and CPE-110) mutants were crossed with wildtype and F1 progeny were generated. The $F_1$ progeny from these crosses exhibited a wild type phenotype indicating that the traits are governed by recessive alleles.

## Sequencing of mutant lines

We performed whole genome re-sequencing of the wild type (three genetic stocks) and eight mutant lines showing different traits to study the extent of mutagenesis. We obtained high quality sequences with an average depth of 41x (Table 3). Greater than 98% of the reads from the mutant lines mapped to the Samba Mahsuri reference genome (Table 3). Analyses of the EMS-induced SNPs revealed that the dissimilarity between Samba Mahsuri and the mutant lines ranges from 0.008% to 0.032% (Table 4). The SNPs/Mb was ranging from 76.3 to 322.3 bases (Table 4). As expected, many more SNPs were present in non-coding regions as compared to the coding regions (S13 Table). The high yield mutant TI-128 had the highest number of SNPs whereas the TI-42 line (tolerant to BLB) had the lowest number of SNPs in both coding and non-coding regions. Annotation of the SNPs showed that 8.86 to 10.01% of the SNPs were in the coding regions of the mutant lines (S13 Table) and relatedness analysis revealed

**Table 3. Sequence statistics of Samba Mahsuri (wild type) and mutant lines with different traits.**

| S. No. | Line | Type or trait of Line | Number of Raw Reads (R1+R2) | QC Passed Reads (R1+R2) | Sequen ced bases (Gb) | Cover age (X) | Mapped QC passed Reads (R1+R2) | Mapp ed Reads (%) | Reference used |
|---|---|---|---|---|---|---|---|---|---|
| 1 | SM-E | Wild type | 112285084 | 102241208 | 15.34 | 39.22 | 100885182 | 98.67 | **R498** |
| 2 | SM-G | Wild type | 110937222 | 111916395 | 16.79 | 42.94 | 110657624 | 98.88 | |
| 3 | SM-C | Wild type | 123573016 | 111003694 | 16.65 | 42.59 | 122513940 | 98.27 | |
| 4 | TI-109 | Complete Panicle Exsertion | 114514030 | 114191395 | 17.13 | 43.81 | 103577844 | 98.73 | **Samba Mahsuri** |
| 5 | TI-110 | Complete Panicle Exsertion | 107814208 | 96892528 | 14.53 | 37.17 | 96892528 | 98.77 | |
| 6 | TI-128 | High Yield; High Spikelets per Panicle | 113249162 | 102101092 | 15.32 | 39.17 | 102101092 | 98.89 | |
| 7 | TI-35 | Elongated Uppermost Internode | 108635160 | 108364345 | 16.25 | 41.57 | 97603326 | 98.76 | |
| 8 | TI-38 | Complete Panicle Exsertion | 113152500 | 102614244 | 15.39 | 39.37 | 102614244 | 98.86 | |
| 9 | TI-42 | Bacterial Blight Tolerance | 106403198 | 78348650 | 11.75 | 30.06 | 78348650 | 98.62 | |
| 10 | TI-50 | High Yield; High Spikelets per Panicle | 109924526 | 99290257 | 14.89 | 38.09 | 99290257 | 98.90 | |
| 11 | SB-6 | Sheath Blight Tolerance | 126418290 | 125999511 | 18.90 | 48.34 | 112679044 | 98.62 | |

clustering of the mutant lines (S1 Fig). The number of SNPs in the CDs ranged from 2962 (TI-42) to 12268 (TI- 128). The number of SNPs in the 3' UTR region were ranging from 1195 (TI-35) to 4855 (TI- 128) whereas the number of SNPs in the 5' UTR region ranged from 1147 (TI-42) to 4320 (TI- 128). The number of SNPs in the non-coding regions ranged from 24301 (TI-42) to 103796 (TI- 128).

## Discussion

Samba Mahsuri (BPT-5204) is a popular rice variety in India because of its excellent cooking and eating qualities but it is highly susceptible to many biotic and abiotic stresses. The everlasting demand for this variety has encouraged various researchers to select this variety for marker assisted selection (MAS) to improve tolerance for some of the stresses, where the donors are

**Table 4. Details of SNPs in the mutant lines with respect to wild type.**

| S. No. | Line | Trait | Number of SNPs | Dissimilar ity index (%)a | SNPs per Mb |
|---|---|---|---|---|---|
| 1 | TI-109 | Complete Panicle Exsertion | 69765 | 0.018 | 178.4 |
| 2 | TI-110 | Complete Panicle Exsertion | 59320 | 0.015 | 151.7 |
| 3 | TI-128 | High Yield; High Spikelets per Panicle | 126019 | 0.032 | 322.3 |
| 4 | TI-35 | Elongated Uppermost Internode | 34121 | 0.009 | 87.3 |
| 5 | TI-38 | Complete Panicle Exsertion | 65556 | 0.017 | 167.7 |
| 6 | TI-42 | Bacterial Blight Tolerance | 29850 | 0.008 | 76.3 |
| 7 | TI-50 | High Yield; High Spikelets per Panicle | 124933 | 0.032 | 319.5 |
| 8 | SB-6 | Sheath Blight Tolerance | 37387 | 0.010 | 95.6 |

available in the gene pool. To enhance the scope of MAS and to develop the donors for traits where the donors are not available, creation of variation through mutagenesis was sought. Among various physical and chemical mutagens, EMS majorly induces point mutations and less chromosomal segment breaks, as well demonstrated in rice [9]. The screening of mutants generated by T-DNA or transposons suffer from restrictions for field evaluation as they are considered as genetically modified (GM) but this restriction does not apply for the EMS induced mutants. Owing to these advantages, in the present study, EMS was opted for induction of mutagenesis. Till date, more than 22 mutant resources were developed in diverse genetic backgrounds of rice using different mutagens across the globe [48], but no studies were reported on the development of mutagenic population in the background of Samba Mahsuri.

Identification of suitable mutagenic dose is the key component in developing mutagenic populations since it determines the mutation densities. To generate such population, Samba Mahsuri seed were treated with EMS in two different concentrations (1.2% and 0.8% for 12 hrs) and the 1.2% treatment yielded a greater frequency of mutants with favored agro-morphological traits. Similar mutation induction studies were conducted by Mohapatra et al. [10] in *aus* cultivar Nagina 22 and confirmed that 1.5% for 12 hrs of EMS treatment generated useful mutants after using a wide range of EMS concentrations (0.2–2%). Wu et al. [49] established a mutant population in IR64 using five different EMS concentrations (0.4, 0.6, 0.8, 1.0 and 1.6%) for kill curve analysis and observed that the 1.6% concentration exhibited the highest frequency rate of mutations. Talebi et al. [50] used six different concentrations (0.25%, 0.50%, 0.75%, 1%, 1.25%, 1.5% and 2%) of EMS to treat MR219, a Malaysian *indica* variety and fixed 0.25% and 0.50% for 12 hrs as LD values for generating variation. Several mutagenesis studies on plants have reported that the optimal concentration of the chemical mutagen can vary with the variety used, which might be due to possible influence of the genome on the effects of the mutagen [51].

The progeny from a set of 10,500 Samba Mahsuri $M_1$ mutant lines were characterized for agro-morphological traits and biotic stresses from $M_2$ to $M_5$ generations and mutants were identified that have variations in different morphological characters like height (tall, dwarf, bonsai) and pink apiculus. Similar plant height variations were also identified by Mohapatra et al. [10]. Our observations also suggested that some mutants might have mutations with large phenotypic effects which fell well outside the phenotype of the wild type. The flag leaf and top leaf blades play a major role in photosynthesis and affect the grain yield of rice [52]. Therefore, we classified mutants having broad and long flag leaf variations in the category of physiological traits. These mutants might carry mutations among alleles/genes responsible for flag leaf development. Earlier studies demonstrated that a mutation in the Flag Leaf gene (*NAL 1*) affects various yield related traits in rice [53]. Similarly, Sakamoto et al. [54] explained that the brassino-steroid deficient mutant having erect leaves showed increase in biomass and grain yield in rice. Phenotypes such as albino, xantha and coloration mutants were observed in $M_1$ generation, but they were not observed in adult plants of $M_2$ generation indicating that they were lethal.

Improvement of lodging resistance by incorporation of the semi-dwarf trait alone is not sufficient and other traits such as strong culm are needed for reinforcement of the phenotype [55]. According to Ookawa et al. [56], the diameter and wall thickness will influence the parameter of the physical strength of the culm. A QTL associated with strong culm was earlier identified by Ookawa et al. [57]. Using chromosomal segment substitution lines, they have reported that Strong Culm 2 (*SCM 2*), a gain-of-function mutant of *APO1*, was responsible for the strong culm phenotype. The strong culm mutants identified in this work need to be characterized further to identify the genes/alleles that are contributing to this trait. Maturity

duration is one of the important growth parameters that determine crop productivity under a given climatic condition. We recovered thirty-six early maturing mutants (maturing between 108–135 days). Mutants that are similar in yield with wild type but having an early maturing nature are very useful for the rice varietal improvement programme. Similarly, Chakrabarti et al. [57] developed early maturing PNR series mutants utilizing gamma radiations in Basmati 370 PNR-17-3 and Mustikarini et al. [58], developed a gamma-ray induced red rice (*Celak Madu* accession) mutant which matures earlier than the wild type (i.e., in 115 days) and is drought tolerant.

The stay green or delayed leaf senescence phenotype of the plant is another attractive and desired characteristic of rice. Very few genetic sources are available for this trait and hence the stay green mutants identified in this study may be helpful in improving rice production under stressful environment conditions. In this connection it is pertinent to note that Ramkumar et al. [59] have identified a stay green mutant (SGM-3) of Nagina22 which has better harvest index under drought stress.

Similarly, elongation of the uppermost internode (EUI) is an important component of plant architecture that can be used to overcome the sheathed panicle of the rice male sterile line [60] by enhancing pollen dispersal and facilitating hybrid rice seed production [61]. In our study, the identified EUI mutants were observed to have better upper node elongation than the wild type and their incorporation into the CMS line can help enhance pollen dispersal. Okuno and Kawai [62] first reported EUI mutants, derived from the Japanese rice cultivar Norin 8 by gamma-ray treatment. Similar EUI studies were conducted by Rutger and Carnahan [63] and they reported enhanced upper internode elongation during the heading stage. Similarly, incomplete panicle emergence leads to yield reduction [64] which is generally observed in Samba Mahsuri resulting in 10–15% reduction in yield. Hence, the identified mutants that have complete panicle emergence may play an important role in rice breeding programmes to increase yield of Samba Mahsuri as well as other rice varieties whose yield may be affected due to incomplete panicle emergence.

A number of high yielding mutant lines were identified that have more tiller numbers and higher grain numbers. Similarly, Roy et al. [65] identified eight high yielding aromatic mutants in the background of Tulpanji using gamma irradiation. Soomro et al. [66] identified two high yielding mutant lines derived from IR6, through gamma rays. Several high yielding mutants were also popular in many countries such as Vietnam (55 mutant varieties), Bangladesh (44 mutant varieties) and Thailand (two aromatic mutant varieties (RD6, RD15) where they have been cultivated over millions of hectares [67]. In India, the PNR-381 and PNR- 102 aromatic rice varieties are early maturing rice mutants that are derived from the Basmati 370 PNR-17-3 cultivar and have become popular in Haryana and Uttar Pradesh States (https://mvd.iaea.org) [57].

In the present study, different grain type mutants were isolated. They could have importance in varietal improvement since the visual characteristics of rice grains is an important attribute that affects consumer's preference. Azad et al. [68] developed different grain type mutants by irradiating IR8 with gamma rays. Different grain types (short, long, bold) were also reported from the Nagina22 mutant population [10]. A mutant line derived from Kitake rice variety using fast-neutron (FN) mutagenesis carried a new grain shape 9–1 (*gs9–1*) allele that also affects grain size [69].

In this study, mutants having variations in multiple traits were observed and such kinds of mutants were also reported earlier in Nagina 22 background [10]. These mutants may either carry point mutations in multiple genes that independently affect the observed traits or carry a mutation that has a pleiotropic phenotype [9, 70].

In the present study mutation frequency of many mutant types showed a decrease in the number as we progressed from $M_2$ to $M_5$. We are unable to explain this apparent reversion phenomenon as the phenotype was quite apparent in the $M_2$ generation. One possibility is that there are some physiological changes, such as epigenetic changes, that arise due to mutagenesis, and which revert back to the wild type phenotype/physiological state in subsequent generations. One example of such a reversion was the sickle shape seed phenotype in $M_2$ generation which had reverted back to wild type seed shape by the $M_5$ generation.

Biotic stresses (insect pests and diseases) cause more than 42% of crop yield reduction in the world [71]. Among the biotic stresses, Yellow stem borer, Sheath blight and BLB count amongst the important biotic stresses in rice. The resistance breeding for yellow stem borer has not gained any momentum due to the lack of suitable resistance/tolerance sources [72]. Sheath Blight is another challenging disease due to unavailability of a reliable resistance source [73]. Whereas, for management of BLB, there is an emergent need to widen the repertoire of resistance genes to enhance the spectrum and durability of resistance to *Xanthomonas oryzae* pv. *oryzae*. Mutagenesis in a gene may result in loss or gain in function with varying levels of expression and such types of mutations have prime importance while dealing with biotic stresses. Therefore, we attempted to identify resistance/tolerance stocks for YSB, Sheath blight and BLB from the mutagenized population. In a similar mutation study conducted by Wu et al. [49], by irradiating IR64 with gamma rays, rice mutants that are resistant to Blast, BLB and tungro disease were identified. Similarly, Mohapatra et al. [10] identified BB resistance mutants in an EMS induced Nagina 22 population. In the present study the identified biotic stress tolerance mutants under multi-location trials showed stability and durability of the mutants across different locations of India. Our study is consistent with the above-mentioned studies which indicate that mutagenized populations harbor a great amount of variability that can be recovered if the mutant population is subjected to appropriate screening techniques.

Earlier studies reported the frequency of EMS induced mutations within a range of 900–2000 and 2163–25753 in rice [42, 48]. Our study revealed 2962–12268 SNPs in the coding sequences (CDS). The exploitation of identified large number of SNPs by developing $F_2$ mapping populations can lead to identification of mutations responsible for the trait of interest. Evaluation of dis-similarity of the mutant lines with wild type using whole genome sequence analyses indicated that the mutants have diversity which ranged from 0.008% to 0.032% that indicate a range of variation.

Recent advances in high throughput sequencing technologies and computational approaches have facilitated the mapping of causal mutations in mutant lines [74, 75]. In addition to gene identification, an allelic series comprised of SNPs and INDELs can be useful in assessing gene function and for practical purposes of crop improvement. Therefore, comparison among the allelic series across mutants and natural germplasm could help to infer the functional polymorphisms related to phenotypes observed in mutants and germplasm.

The mutants identified in this screen, especially those that have agronomically important phenotypes with significant similarity to the wild type could be useful to use in MutMap and QTL Seq for the identification of linked SNPs. Such studies will generate new knowledge and also have the potential for application in crop improvement.

## Conclusions

Exploiting genetic variability induced through mutations can be a powerful strategy in rice improvement where genetic gains may be diminishing due to a narrow gene pool. To generate variability, we have produced a large Samba Mahsuri mutant population that can be further employed in crop improvement. The huge demand for this variety and its wider adaptability and

combining ability makes this mutant population a very useful resource for rice improvement. The value of these stocks will be enhanced with increasing usage and extensive screening under a wide range of conditions. Systematic screens and mapping studies will not only help in generating new knowledge but will also help in translating the knowledge for crop improvement.

## Supporting information

**S1 Table. Primer sequence of all tested genes to detect allelic variations.**
(XLS)

**S2 Table. List of mutants that exhibit variations in morphological traits from $M_2$ to $M_5$ generations.**
(XLSX)

**S3 Table. List of mutants that exhibit variations in physiological traits from $M_2$ to $M_5$ generations.**
(XLSX)

**S4 Table. List of mutants that exhibit variations in panicle type from $M_2$ to $M_5$ generations.**
(XLSX)

**S5 Table. List of mutants that exhibit variations in panicle emergence from $M_2$ to $M_5$ generations.**
(XLSX)

**S6 Table. List of mutants that exhibit variations in yield attributing traits from $M_2$ to $M_5$ generations.**
(XLSX)

**S7 Table. List of mutants that exhibit variations in grain types from $M_2$ to $M_5$ generations.**
(XLSX)

**S8 Table. List of mutants that exhibit tolerance for YSB from $M_2$ to $M_5$ generations.**
(XLS)

**S9 Table. List of mutants that exhibit tolerance for sheath blight from $M_2$ to $M_5$ generations.**
(XLS)

**S10 Table. List of mutants that exhibit resistance for bacterial blight from $M_2$ to $M_5$ generations.**
(XLS)

**S11 Table. Screening of selected mutants with multiple isolates under multiple locations.**
(XLS)

**S12 Table. List of mutants that exhibit variations in multiple traits.**
(XLS)

**S13 Table. Distribution of SNPs among different regions of gene of sequenced mutant lines.**
(XLS)

**S1 Fig. Related analysis indicated a high similarity among the mutant lines which is seen as single cluster containing majority of the mutant lines.**
(TIF)

## Author Contributions

**Conceptualization:** Ramesh Venkata Sonti, Maganti Sheshu Madhav.

**Data curation:** Gopi Potupureddi, Suneel Ballichatla, Gokulan C. G., Swathi Lekkala, Karteek Jallipalli, Gayathri M. G., Ershad Mohammad, Milton M, Srikanth Arutla, Rajender Burka, Padmakumari Ayyangari Phani, Kranthi Brahma, Raju Madnala.

**Formal analysis:** Gopi Potupureddi, Vishalakshi Balija, Suneel Ballichatla, Karteek Jallipalli, Gayathri M. G., Laha Gouri Shankar, Padmakumari Ayyangari Phani, SubbaRao Lella Venkata, Sundaram Raman Meenakshi, Kranthi Brahma.

**Funding acquisition:** Hitendra Kumar Patel, Ramesh Venkata Sonti, Maganti Sheshu Madhav.

**Investigation:** Suneel Ballichatla, Swathi Lekkala, Karteek Jallipalli, Laha Gouri Shankar, Padmakumari Ayyangari Phani, SubbaRao Lella Venkata, Sundaram Raman Meenakshi, Maganti Sheshu Madhav.

**Methodology:** Gopi Potupureddi, Vishalakshi Balija, Suneel Ballichatla, Karteek Jallipalli, Gayathri M. G., Ershad Mohammad, Milton M, Rajender Burka, Laha Gouri Shankar, Padmakumari Ayyangari Phani, Sundaram Raman Meenakshi, Raju Madnala, Maganti Sheshu Madhav.

**Project administration:** Komal Awalellu, Viraktamath B. C., Ravindra Babu Vemuri, Ramesh Venkata Sonti.

**Software:** Gokulan C. G.

**Supervision:** Laha Gouri Shankar, Padmakumari Ayyangari Phani, SubbaRao Lella Venkata, Sundaram Raman Meenakshi, Viraktamath B. C., Ravindra Babu Vemuri, Hitendra Kumar Patel, Maganti Sheshu Madhav.

**Validation:** Vishalakshi Balija, Suneel Ballichatla, Karteek Jallipalli, Kranthi Brahma.

**Visualization:** Suneel Ballichatla, Gokulan C. G., Komal Awalellu, Karteek Jallipalli, Milton M, Laha Gouri Shankar.

**Writing – original draft:** Ramesh Venkata Sonti, Maganti Sheshu Madhav.

**Writing – review & editing:** Ramesh Venkata Sonti, Maganti Sheshu Madhav.

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
