## [Decision Letter · Decision Letter 0]

29 Jul 2021

PONE-D-21-18858

Mutation resource of Samba Mahsuri revealed the presence of high extent of variations among key traits for rice improvement

PLOS ONE

Dear Dr. Sheshu Madhav,

Thank you for submitting your manuscript to PLOS ONE. After careful consideration, we feel that it has merit but does not fully meet PLOS ONE’s publication criteria as it currently stands. Therefore, we invite you to submit a revised version of the manuscript that addresses the points raised during the review process.

Two reviewers suggested minor revisions of  the main Text, Table and Figures of the manuscript. All these comments/ suggestions given by reviewers are relevant and important for improving the overall quality of the manuscript. Authors should modify and revise the manuscript according to the comments of both reviewers. The manuscript develops a novel genetic resource (mutant populations) for the rice community which can be used for translational genomics to drive crop improvement in rice. Therefore, the manuscript is new and most vital for consideration of publication in the journal.     

We look forward to receiving your revised manuscript.

Kind regards,

Swarup Kumar Parida, Ph.D.

Academic Editor

PLOS ONE

“Council of Scientific and Industrial Research (CSIR), New Delhi, India and ICAR -IIRR”

“NO authors have competing interests”

 This information should be included in your cover letter; we will change the online submission form on your behalf

5. Thank you for stating the following in the Funding Section of your manuscript:

“The authors acknowledge the Council of Scientific and Industrial Research (CSIR), New Delhi, India and ICAR -IIRR for providing funds and infrastructure for carrying out the research work. RVS is supported by a J C Bose Fellowship from the Science and Engineering Research Board, Government of India.”

We note that you have provided funding information that is not currently declared in your Funding Statement. However, funding information should not appear in the Funding section or other areas of your manuscript. We will only publish funding information present in the Funding Statement section of the online submission form.

 “Council of Scientific and Industrial Research (CSIR), New Delhi, India and ICAR -IIRR”

6.Please review your reference list to ensure that it is complete and correct. If you have cited papers that have been retracted, please include the rationale for doing so in the manuscript text, or remove these references and replace them with relevant current references. Any changes to the reference list should be mentioned in the rebuttal letter that accompanies your revised manuscript. If you need to cite a retracted article, indicate the article’s retracted status in the References list and also include a citation and full reference for the retraction notice.

Reviewers' comments:

Reviewer's Responses to Questions

**Comments to the Author**

1. Is the manuscript technically sound, and do the data support the conclusions?

Reviewer #1: Yes

Reviewer #2: Yes

2. Has the statistical analysis been performed appropriately and rigorously? 

Reviewer #1: Yes

Reviewer #2: Yes

3. Have the authors made all data underlying the findings in their manuscript fully available?

Reviewer #1: Yes

Reviewer #2: No

4. Is the manuscript presented in an intelligible fashion and written in standard English?

Reviewer #1: Yes

Reviewer #2: Yes

5. Review Comments to the Author

Reviewer #1: To

Chief Editor

PLOS ONE

Dear Sir,

I have made some of the following remarks about the manuscript entitled “Mutation resource of Samba Mahsuri revealed the presence of high extent of variations among key traits for rice improvement”.

1. This article is contributable to science & technology if improvements can be made.

2. As the author is concerned about efficient development of genetic stock through mutagenesis.

3. Title as well as the objectives of the research article are nicely reflected the manuscript.

4. Authors need to follow the exact format of the POLS ONE journal in the text as well as in the reference sections of the manuscript.

5. References should be thoroughly checked and written in the text as well as in the References section as per Journal.

6. Paper required a revision to follow Journal format.

With regards,

R. K. Salgotra

Professor & Coordinator

School of Biotechnology

SKUAST-J, Chatha, Jammu (J&K)

INDIA

Reviewer #2: i. In Table S6, please provide comparative data of mutants at M4 or M5 generation with the wild type for the trait, high grain number using statistical parameters.

ii. In Table S8, provide the YSB score of mutants at M4 generation with wild type.

iii. In Table S12, provide comparative data of mutants at M4 or M5 generation with the wild type for the trait, yield/plant using statistical parameters.

iv. Check English language

6. PLOS authors have the option to publish the peer review history of their article (what does this mean?). If published, this will include your full peer review and any attached files.

Reviewer #1: **Yes: **R K Salgotra

Reviewer #2: No

---

## [Author Response · Author response to Decision Letter 0]

26 Aug 2021

Response to Editor comments:

Thank you for considering our manuscript to PLOS ONE. we submitted the revised version of the manuscript that addresses the points raised by reviewers. Related to financial discloser now the funding agency had no role in study design, data collection and analysis, decision to publish, or preparation of the manuscript

Response to Reviewers

Thank you for giving us the opportunity to submit a revised draft of the manuscript “Mutation resource of Samba Mahsuri revealed the presence of high extent of variations among key traits for rice improvement” for publication in the Plos One journal. We appreciate the time and effort that you and the reviewers dedicated to providing feedback on our manuscript and are grateful for the insightful comments on and valuable improvements to our paper. We have incorporated the suggestions made by the reviewers. 

Reviewers’ comments to the authors:

Reviewer #1:

1. This article is contributable to science & technology if improvements can be made.

Reply: Thank you. Authors improved the manuscript to meet the journal standards.

2. As the author is concerned about efficient development of genetic stock through mutagenesis.

Reply: Yes we are very much concerned to develop lines possessing biotic stresses (YSB, ShB and BLB) tolerance as well as several economically important traits through mutagenesis. These unique genetic resources can be utilized for the identification of novel genes/alleles for different traits. The lines which have better agronomic features can be used as pre-breeding lines.

3. Title as well as the objectives of the research article are nicely reflected the manuscript.

Reply: Thank you. We appreciate the time and effort that you and the reviewers have dedicated to providing your valuable feedback on our manuscript

4. Authors need to follow the exact format of the POLS ONE journal in the text as well as in the reference sections of the manuscript.

Reply: Thank you for this suggestion; we have edited the manuscript and reference in PLOS ONE journal format. 

5. References should be thoroughly checked and written in the text as well as in the References section as per Journal.

Reply: Thank you for pointing this out. We have checked all the reference and revised to journal format

6. Paper required a revision to follow Journal format.

Reply: We agree with this comment. We revised the manuscript in PLOS ONE journal format

Reviewer #2: 

i. In Table S6, please provide comparative data of mutants at M4 or M5 generation with the wild type for the trait, high grain number using statistical parameters.

Reply: Thank you for valuable suggestion. We performed the statistical analysis (ANOVA) of grain number of mutant and wild type and the data is mentioned in Table S6

ii. In Table S8, provide the YSB score of mutants at M4 generation with wild type.

Reply: Thank you for this suggestion. YSB phenotypic score of wild type (Samba Mahsuri) was mentioned at M4 generation 

iii. In Table S12, provide comparative data of mutants at M4 or M5 generation with the wild type for the trait, yield/plant using statistical parameters.

Reply: As suggested by the reviewer the statistical analysis (ANOVA) was performed for yield/plant and the data is mentioned in Table S12.

iv. Check English language

Reply: Thank you for the valuable suggestion. For improvement of English, we have given manuscript to our institute science editor and improved substantially

Thank you,

With regards

Dr. M. Sheshu Madhav

---

## [Editor Report · Decision Letter 1]

6 Oct 2021

Mutation resource of Samba Mahsuri revealed the presence of high extent of variations among key traits for rice improvement

PONE-D-21-18858R1

Dear Dr. Sheshu Madhav,

We’re pleased to inform you that your manuscript has been judged scientifically suitable for publication and will be formally accepted for publication once it meets all outstanding technical requirements.

Kind regards,

Swarup Kumar Parida, Ph.D.

Academic Editor

PLOS ONE
---

## [Editor Report · Acceptance letter]

11 Oct 2021

PONE-D-21-18858R1 

Mutation resource of Samba Mahsuri revealed the presence of high extent of variations among key traits for rice improvement 

Dear Dr. Madhav:

I'm pleased to inform you that your manuscript has been deemed suitable for publication in PLOS ONE. Congratulations! Your manuscript is now with our production department. 

Kind regards, 

on behalf of

Dr. Swarup Kumar Parida 

Academic Editor

PLOS ONE